# A three enzyme system to generate the *Strychnos* alkaloid scaffold from a central biosynthetic intermediate

Evangelos C. Tatsis [1], Inês Carqueijeiro [2], Thomas Dugé de Bernonville [2], Jakob Franke[1], Thu-Thuy T. Dang[1], Audrey Oudin[2], Arnaud Lanoue[2], Florent Lafontaine[2], Anna K. Stavrinides [1], Marc Clastre[2], Vincent Courdavault [2] & Sarah E. O'Connor [1]

Monoterpene indole alkaloids comprise a diverse family of over 2000 plant-produced natural products. This pathway provides an outstanding example of how nature creates chemical diversity from a single precursor, in this case from the intermediate strictosidine. The enzymes that elicit these seemingly disparate products from strictosidine have hitherto been elusive. Here we show that the concerted action of two enzymes commonly involved in natural product metabolism—an alcohol dehydrogenase and a cytochrome P450—produces unexpected rearrangements in strictosidine when assayed simultaneously. The tetrahydro-β-carboline of strictosidine aglycone is converted into akuammicine, a *Strychnos* alkaloid, an elusive biosynthetic transformation that has been investigated for decades. Importantly, akuammicine arises from deformylation of preakuammicine, which is the central biosynthetic precursor for the anti-cancer agents vinblastine and vincristine, as well as other biologically active compounds. This discovery of how these enzymes can function in combination opens a gateway into a rich family of natural products.

[1] John Innes Centre, Department of Biological Chemistry, Norwich Research Park, Norwich NR4 7UH, UK. [2] Université François-Rabelais de Tours, EA2106 Biomolécules et Biotechnologies Végétales, Parc de Grandmont 37200 Tours, France. Evangelos C. Tatsis and Inês Carqueijeiro contributed equally to this work. Correspondence and requests for materials should be addressed to V.C. (email: vincent.courdavault@univ-tours.fr) or to S.E.O. (email: sarah.oconnor@jic.ac.uk)

The medicinal plant *Catharanthus roseus* synthesizes a variety of biologically active monoterpene indole alkaloids, most notably the anti-cancer agents vinblastine and vincristine. Vinblastine and vincristine, along with many other monoterpene indole alkaloids, are derived from the biosynthetic intermediate, preakuammicine, as evidenced by extensive feeding studies and isolation of biosynthetic intermediates from plant material[1, 2]. However, discovery of the enzymes that are responsible for biosynthesis of the preakuammicine skeleton has proven elusive.

Nearly all monoterpene indole alkaloid pathways begin with deglycosylation of strictosidine by a dedicated glucosidase, strictosidine glucosidase (SGD)[3]. Strictosidine aglycone, which can exist in numerous isomers such as 4,21-dehydrogeissoschizine, is then enzymatically transformed into the various monoterpene indole alkaloid skeletons. However, the enzymes responsible for the synthesis of preakuammicine, or any preakuammicine-derived product from strictosidine aglycone remain unknown. While the enzymatic reactions remain cryptic, extensive isolation of reaction intermediates as well as model chemical studies suggest a sequence of reactions: 4,21-dehydrogeissoschizine (strictosidine aglycone isomer) is reduced to geissoschizine, which is followed by oxidative rearrangement and reduction to preakuammicine[4–10]. Preakuammicine can non-enzymatically deformylate to produce the alkaloid akuammicine[8, 10–12] or serve as a precursor for additional alkaloid scaffolds[8].

Here we use a transcriptomic database for the medicinal plant *Catharanthus roseus* to identify two enzymes, an alcohol dehydrogenase (GS) and a cytochrome P450 (GO), that act in tandem on 4,21-dehydrogeissoschizine (strictosidine aglycone) to yield akuammicine, a *Strychnos* alkaloid. These enzymes are characterized by both in vitro and in planta silencing approaches. Akuammicine is known to arise from deformylation of preakuammicine[8, 10–12], which is the central biosynthetic precursor for the anti-cancer agents vinblastine and vincristine, as well as hundreds of other biologically active compounds. Therefore, these enzymes open a gateway into a rich family of natural products, and will facilitate metabolic engineering efforts to make high-value monoterpene indole alkaloids.

## Results

**Identification of gene candidates**. To identify genes responsible for generating *Strychnos* scaffolds such as preakuammicine and akuammicine (Fig. 1a), we mined a transcriptomic database of *C. roseus* tissues[13, 14], which also included data from plants with increased alkaloid levels due to folivory[15]. Gene candidates identified were initially screened in *C. roseus* by virus-induced gene silencing (VIGS), a transient silencing system in which qualitative perturbations in the metabolic profile between silenced and control tissue provide clues to the metabolic function of the gene of interest. Transformation into preakuammicine requires both reduction and oxidation steps (Fig. 1b, Supplementary Fig. 1)[4–10]. Therefore, we focused on alcohol dehydrogenases (reductase) and cytochromes P450 (oxidase); alcohol dehydrogenases have been implicated in other monoterpene indole alkaloid biosynthetic steps[16–18], and cytochromes P450 are well known to play important roles in oxidative transformations in plant metabolism[16, 19].

**Fig. 1** Monoterpene indole alkaloid biosynthesis. **a** All *ca.* 2000 monoterpene indole alkaloids are derived from strictosidine, which is converted to preakuammicine through an unknown series of enzymatic reactions. Preakuammicine is hypothesized to be the precursor for many structurally divergent alkaloids. **b** A schematic overview of the proposed pathway that generates the *Strychnos* alkaloid akuammicine. Strictosidine is deglycosylated by the known enzyme, strictosidine glucosidase (SGD). An isomer of strictosidine aglycone, 4,21-dehydrogeissoschizine, can be reduced to form geissoschizine, which feeding studies suggest rearranges to form preakuammicine. Preakuammicine can deformylate to form the *Strychnos* alkaloid akuammicine, and is also hypothesized to be the precursor for many downstream alkaloids. See Supplementary Fig. 1 for a more detailed picture of monoterpene indole alkaloid biosynthesis

**In planta and in vitro assay of enzyme candidates.** Prioritization of genes through Pearson's correlation coefficient (PCC) determination revealed a limited number of candidates displaying correlation with *SGD* (549 with PCC > 0.6), with only two candidates predicted to encode cytochromes P450 (Supplementary Data 1). One was secologanin synthase, which is involved in upstream monoterpene indole alkaloid biosynthesis[20, 21] and the second was CYP71D1V1, which was previously deposited as JN613015 in a previous report, though not functionally characterized[22]. Silencing *CYP71D1V1* led to a statistically significant

decreases in the production of the alkaloids catharanthine, vindoline and vindorosine (Supplementary Fig. 2a), iboga and aspidosperma type alkaloids that are derived from pre-akuammicine (Supplementary Fig. 1)[23].

All literature reports suggest geissoschizine (*m/z* 353), a reduction product of the 4,21-dehydrogeissoschizine isomer of strictosidine aglycone, is the precursor for preakuammicine (Fig. 1b)[4, 5, 10]. As part of an effort to characterize alcohol dehydrogenases of *C. roseus*, we noted that silencing of one medium chain alcohol dehydrogenase gene (named *GS1* for

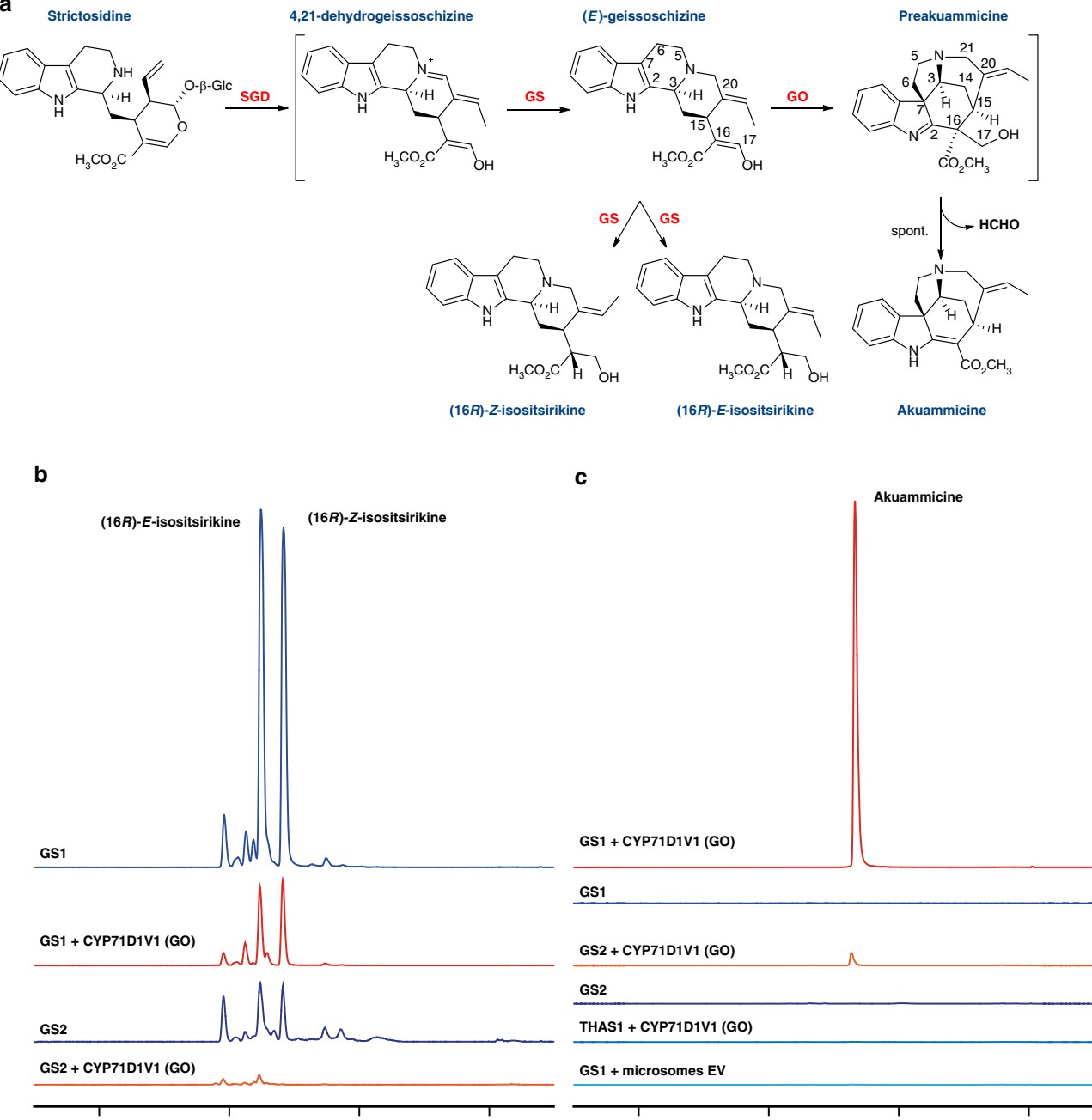

**Fig. 2** Enzymatic activity of GS1 and GS2 and GO (CYP71D1V1). **a** Reaction of strictosidine with SGD, GS and GO. **b** Multiple reaction monitoring (MRM) on a LCMS showing compounds at *m/z* 355 in enzymatic assays of GS1/2 and GO. Isolation and NMR characterization of the two major compounds indicated that they are (16*R*)-*E*-isositsirikine and (16*R*)-*Z*-isositsirikine (Supplementary Figs 5–16). **c** MRM (*m/z* 323) of an akuammicine standard compared with enzymatic products of GS1/2 and GO. THAS1 is a medium chain alcohol dehydrogenase involved in the heteroyohimbine monoterpene indole alkaloid pathway (Supplementary Fig. 1). *EV* empty vector

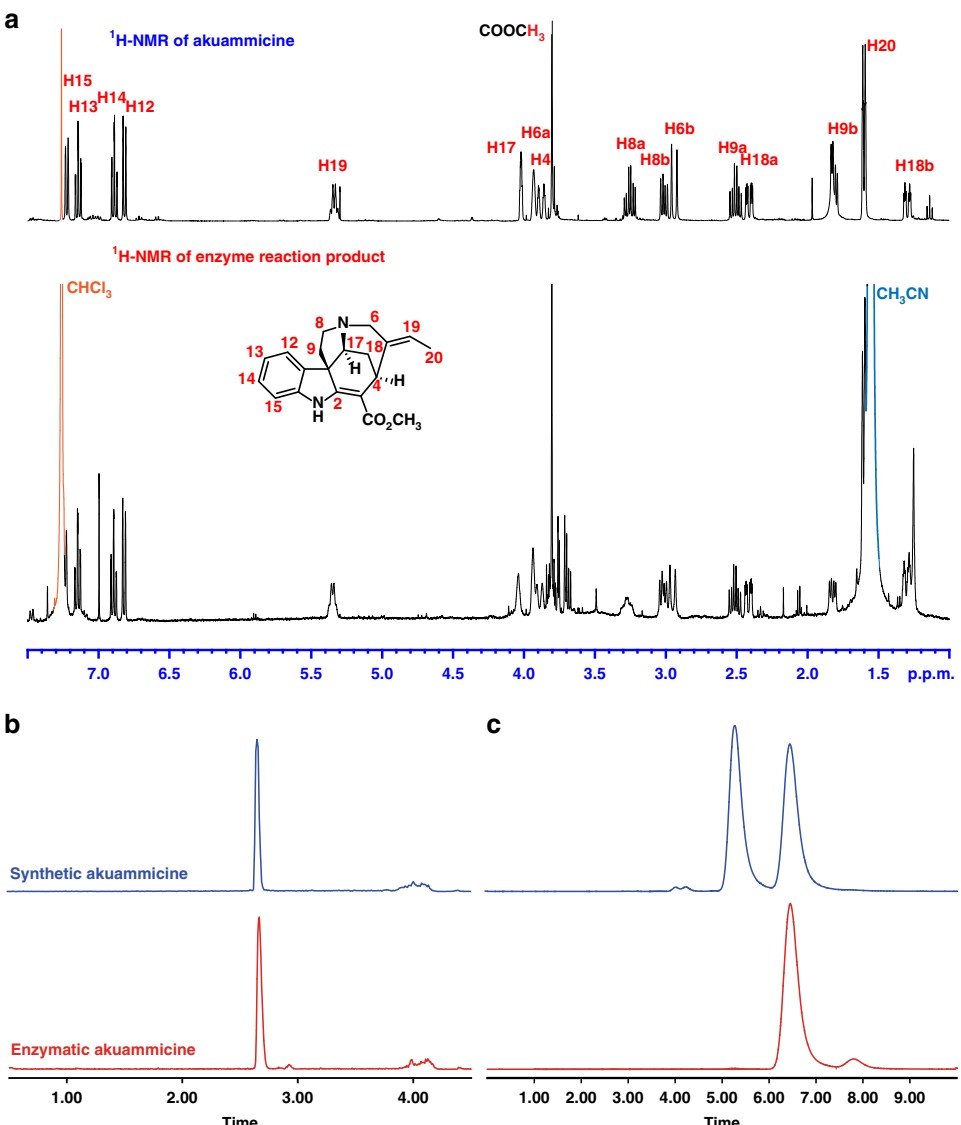

**Fig. 3** Characterization of the GS1/GO product, akuammicine. **a**. [1]H NMR of enzymatically produced akuammicine compared to a synthetic standard (see also Supplementary Figs 19–23). **b**. LC-MS chromatograms with a standard reverse phase column comparing synthetic akuammicine with the enzymatic product. **c** Chiral LC-MS chromatograms comparing the racemic synthetic standard with the enzymatic product

geissoschizine synthase), led to only minor changes in alkaloid profiles (Supplementary Fig. 3). Since VIGS phenotypes can be attenuated by factors such as enzyme redundancy or molecule transport, we nevertheless assayed the protein in vitro. When the protein encoded by this gene (initially reported but not functionally characterized during a search for strictosidine biosynthetic genes as KF302079.1[16]) was heterologously expressed in *E. coli* (Supplementary Fig. 4) and assayed with strictosidine in the presence of SGD, the enzymatic reaction yielded a mixture of products, primarily consisting of two compounds (*m/z* 355) (Fig. 2b). Isolation and NMR characterization showed that the two major products of this enzymatic reaction are (16 *R*)-*E*-isositsirikine (high-resolution MS [M + H]+ 355.2020; isolated in ca. 38% yield) and (16 *R*)-*Z*-isositsirikine (high-resolution MS [M + H]+ 355.2014; isolated in *ca.* 7% yield) (Fig. 1a, Supplementary Figs 5 and 16)[24, 25]. In addition, a homolog of *GS1* (*GS2*, KF302078.1[16]) was also identified and tested. This enzyme was purified in lower yields from *E. coli* (Supplementary Fig. 4) but produced the same products as GS1 in decreased quantity. While the formation of isositsirikine initially

led us to dismiss GS1/GS2 as candidates for geissoschizine synthase, we noted that in *CYP71D1V1* silencing experiments, a compound with the same *m/z* and retention time as (16 *R*)-*E*-isositsirikine accumulated (Supplementary Fig. 2b, Supplementary Fig. 17). This provided an important clue that the enzymatic reactions of GS1 and CYP71D1V1 are coupled. We hypothesized that GS1/GS2 initially produces geissoschizine, which is captured and then oxidized to preakuammicine when the correct downstream oxidase is present. However, in the absence of a downstream enzyme, geissoschizine could be over-reduced by GS1/GS2 to the more stable (16 *R*)-*Z*, *E*-isositsirikine products. Notably, the *E*-isositsirikine isomer, which is the major enzymatic product and also the isomer that accumulates in CYP71D1V1-silenced tissues, has the same C19–20 bond geometry as geissoschizine[9].

To test this hypothesis, CYP71D1V1, a microsomal enzyme, was heterologously expressed in a yeast strain harboring a cytochrome P450 reductase, and the microsomes were harvested. When SGD, CYP71D1V1 and GS1 were incubated together with strictosidine, using selected ion monitoring at *m/z* 355 we noted

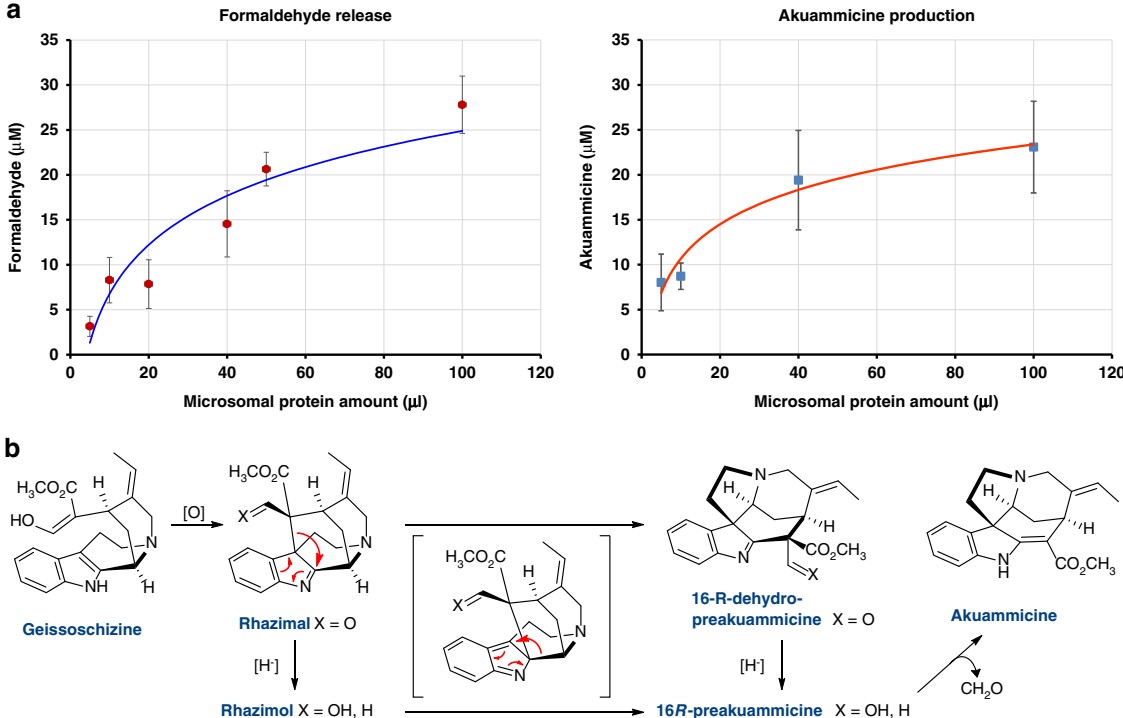

**Fig. 4** Deformylation in the formation of akuammicine. **a** Deformylation as measured with Nash reagent correlates with akuammicine production as measured by mass spectrometry (see also Supplementary Figs 27–29). **b** Possible mechanistic scenario for rearrangement of geissoschizine to the preakuammicine skeleton as suggested by Benayad et al.[10]. Error bars represent standard error of three replicates

the same product profile observed with GS1 lacking CYP71D1V1, but the amounts of the products appeared to be decreased (Fig. 2b). However, no new product corresponding to pre-akuammicine (*m/z* 353), dehydropreakuammicine (*m/z* 351) or any potential isomers of these compounds, was observed. Notably, preakuammicine can non-enzymatically deformylate to yield akuammicine (*m/z* 323)[8], and is generally regarded as the direct precursor to akuammicine[10]. Careful analysis of the mass spectra of the GS1 (or GS2) and CYP71D1V1 reaction revealed a compound with *m/z* 323 (high-resolution MS [M + H]+ 323.1760) (Fig. 2c). This enzymatically produced compound was isolated (ca. 12.5% yield), characterized by NMR (Supplementary Figs 18–23) and shown to be identical to a synthetic standard of akuammicine (Fig. 3a, b)[26]. Moreover, the enzymatic akuammicine product was observed to be enantiopure on a chiral column (Fig. 3c). Formation of akuammicine was absolutely dependent on the presence of CYP71D1V1-expressing microsomes (Fig. 2c). On this basis, CYP71D1V1 was named geissoschizine oxidase (GO). In addition, replacement of GS1 or GS2 with another medium chain alcohol dehydrogenase, THAS1, which is involved in an unrelated branch (heteroyohimbine) of monoterpene indole alkaloid biosynthesis[18] failed to yield akuammicine (Fig. 2c).

GS1 and GO must be added to strictosidine aglycone simultaneously for akuammicine to be produced (Supplementary Fig. 24), suggesting that the GS1 reaction product that acts as the substrate for GO (i.e., geissoschizine) is rapidly metabolized. A small amount of compound with a mass consistent with geissoschizine (*m/z* 353) was observed in the GS1 enzymatic reaction, but this compound could not be isolated in sufficient quantities for characterization (Supplementary Fig. 25). In vivo, both GS1 and GO transcripts are enriched in the epidermis relative to whole leaf, consistent with the epidermal location of other monoterpene indole alkaloid enzymes such as strictosidine synthase and SGD (Supplementary Fig. 26a)[27]. At the subcellular level, YFP-labeling of GO and GS1 suggests that GS1 has a nucleocytosolic location while GO is anchored to the ER, potentially facilitating GS1-reaction product uptake (Supplementary Fig. 26b). Collectively, these data suggest that GS1 and GO are located in close proximity in planta, consistent with a coupled role in vivo.

**Mechanism of akuammicine formation.** The mechanistic basis of the rearrangement of the tetrahydro-β-carboline scaffold of strictosidine to the *Strychnos* scaffold remains to established, but the literature provides some clues. Akuammicine can be generated from preakuammicine by non-enzymatic deformylation (Fig. 4a)[8, 10]. Production of formaldehyde during the enzyme assay in approximately equimolar concentration as akuammicine provides indirect support for the formation of preakuammicine (Fig. 4a, Supplementary Figs 27–29). Geissoschizine, which has been shown by feeding studies to be the akuammicine precursor, is proposed to undergo oxidative reaction to form rhazimal. Recent elegant model chemistry suggests that rhazimal, or its reduced counterpart, rhazimol, can rearrange to dehy-dropreakuammicine or preakuammicine, respectively (Fig. 4b)[10]. Therefore, we hypothesize that a reductase generates geissoschi-zine from strictosidine aglycone (4,21-dehydrogeissoschizine isomer). Then, an oxidase transforms geissoschizine to rhazimal, which then rearranges to dehydropreakuammicine, which is then reduced to preakuammicine. Alternatively, rhazimal could be reduced to rhazimol, which rearranges directly to preakuammicine. However, monoterpene indole alkaloid bio-synthesis in strychnine biosynthesis likely proceeds via dehy-dropreakuammicine[28]. GS1/GS2 or an endogenous reductase present in the yeast microsomes could catalyze the reduction of rhazimal or dehydropreakuammicine.

## Discussion

All monoterpene indole alkaloids emerge from rearrangement of the same precursor, strictosidine[1]. A number of these alkaloids (e.g., vinblastine, strychnine, quinine, camptothecin) are used clinically[1, 2]. The different mechanisms by which strictosidine rearranges to form these structurally diverse compounds have been studied extensively, though only enzymes that catalyze the simplest rearrangement have been reported to date[17, 18]. One of the most important—and cryptic—rearrangements of strictosidine is its conversion into the biosynthetic intermediate preakuammicine, the precursor for the *Strychnos* alkaloids, which include neurotoxic agents such as strychnine, as well as the precursor for the aspidosperma and iboga alkaloids that include the anti-cancer agent vinblastine (Supplementary Fig. 1)[1].

While recent synthetic efforts have demonstrated how the monoterpene indole alkaloids can be chemically synthesized from a common scaffold[29], the enzymes that control rearrangement and chemical diversification of the central intermediate found in the biological system have largely remained cryptic. In this manuscript, we report the discovery of two enzymes, an alcohol dehydrogenase (GS) and a cytochrome P450 (GO), that convert strictosidine aglycone into the *Strychnos* alkaloid akuammicine, a κ-opioid receptor agonist[11, 12]. Together, these three enzymes catalyze a series of tandem reactions that lead to the remarkable rearrangement of the tetrahydro-β-carboline strictosidine substrate into the *Strychnos*-type scaffold. VIGS data are supportive of a role in planta, though the weak phenotype of GS suggests that there may be redundancy with this enzyme.

Though extensive model chemistry reported in the literature has provided some mechanistic insights into this enzymatic transformation, aspects of the mechanism of this transformation remain cryptic. Regardless, the discovery of GO and GS1 provides considerable insight into the biocatalytic processes involved in the diversification of monoterpene indole alkaloid structures, and provides an important step forward for the complete elucidation of metabolic pathways leading to biologically active molecules in this family, such as vinblastine. Importantly, this work has highlighted that to unlock the chemical diversity of certain plant pathways, the instability of the reaction intermediates requires that enzymes must be tested in combination. Therefore, methods to rapidly screen enzyme combinations are likely to be required to identify the genes of many plant natural product pathways. In summary, the discovery of these enzymes demonstrate the extraordinary chemistry that takes place in plants, and this discovery will enable metabolic engineering efforts to overproduce these high-value molecules.

## Methods

**Correlation analysis for selection of P450 candidates**. An expression matrix was generated using the CDF97 consensus transcriptome[30] after pseudo-mapping reads from paired-end sequencing runs available from EBI ENA (ERR1512369, ERR1512370, ERR1512371, ERR1512372, ERR1512373, ERR1512374, ERR1512375, ERR1512376, ERR1512377, SRR1144633, SRR1144634, SRR1271857, SRR1271858, SRR1271859, SRR342017, SRR342019, SRR342022, SRR342023, SRR646572, SRR646596, SRR646604, SRR648705, SRR648707, SRR924147, and SRR924148). Quantification was achieved with Salmon v0.7.2 in the variational bayesian optimized (--vbo) quasi-mapping mode with bias correction (--biasCorrect)[31]. Correlations were calculated with R (R Development Core Team, 2015).

**Cloning**. The entire coding sequence of GS1 was amplified from *Catharanthus roseus* cDNA using primers 5′-AAGTTCTGTTTCAGGGCCCGGCTGGAGAAA CAACAAAACTAG-3′ and 5′-ATGGTCTAGAAAGCTTTATTCCTCAAATTTC AATGTATTT-3′ and cloned into pOPIN-F expression vector (N terminus 6x HisTag) using In-Fusion cloning while GS2 was cloned into the same expression vector using the following synthesized DNA fragment with GeneArt String method (ThermoFischer, USA) 5′-AAGTTCTGTTTCAGGGCCCGGCTGGAGAAACAA CAAAACTAGACCTTTCAGTGAAGGCCATCGGATGGGGTGCTGCAGATGC ATCTGGCGTTCTTCAGCCCATTAGGTTCTATAGAAGAGCCCCTGGTGA ACGGGATGTGAAGATTAGAGTTTTGTACTGTGGTGTGTGCAATTTCGAT

ATGGAAATGGTCAGAAACAAGTGGGGTTTCACTAGATACCCTTATGTTT TTGGACATGAGACCGCCGGTGAGGTGGTAGAAGTTGGGAAGAAAGTAG AGAAATTCAAGGTTGGAGATAAGGTGGGCGTGGGATGTATGGTCGGAT CTTGTGGCAAATGTTTCCATTGTCAAAACGAAATGGAGAATTACTGCC CGGAGCCTAATTTGGCTGATGGATCTACTTACCGTGAAGAAGGAGAAC GTTCCTATGGAGGTTGTTCAAATGTCATGGTTGTTGATGAAAAATTCG TCCTTAGATGGCCCGAAAATTTGCCTCAAGATAAAGGAGTTCCTCTCCT CTGTGCTGGGGTTGTTGTTTATAGCCCCATGAAATATATGGGATTTGAT AAGCCAGGAAAGCATATTGGGGTTTTTGGGTTGGGTGGTCTTGGTTCCA TTGCTGTTAAGTTTATTAAAGCTTTTGGTGGTAAGGCTACTGTTATTAGT ACATCAAGGCGTAAAGAGAAGGAAGCCATTGAAGAGCATGGAGCTGA TGCTTTTGTTGTCAACACTGACTCTGAACAATTGAAGGCTCTGGAAGGT ACTATGGATGGTGTTGTGGACACCACCCCAGGTGGCCGCACTCCTAT GCCACTTATGCTCAATTTGGTTAAGTTTGACGGCGCCGTTATACTCGTC GGTGCACCGGAGACGCTATTTGAGCTCCCGTTGGAGCCTATGGGAAGGA AAAAGATAATCGGAAGCTCCACTGGAGGTCTCAAAGAGTATCAAGAAGT GCTTGATATTGCAGCCAAACACAACATTGTATGTGATACTGAGGTTAT TGGGATTGATTATCTCAGCACTGCTATGGAACGCATCAAGAATTTGGAT GTCAAGTACCGATTCGCGATCGACATTGGGAATACATTGAAATTTGAGG AATAAAGCTTTCTAGACCAT.-3′

The CYP71D1V1 coding sequence was amplified using primers 5′-CTGAGAA CTAGTATGGAGTTTTCTTTCTCCTCACCAG-3′ and 5′-CTGAGAACTAGTCT AATCGTTAACAAGATGAGGAACCAATT-3′ and cloned into pESC-HIS expression vector (Agilent Technologies) prior to protein expression in yeast.

**Protein expression**. GS1 and GS2 enzymes were expressed in SoluBL21 (DE3) *E. coli* cells (Genlantis) grown in LB medium. A starter culture was grown overnight at 37°C in 100 mL of 2xYT media supplemented with carbenicillin (100 μg/ml). A 1:100 dilution in fresh 2xYT (2 × 1 l) media supplemented with antibiotics was prepared and allowed to grow at 37°C to an OD600 of 0.8. Protein production was induced by addition of 0.5 mM IPTG and the cultures were shaken at 18°C for 16 h. Cells were collected by centrifugation, lysed by sonication in Buffer A (50 mM Tris-HCl pH 8, 50 mM glycine, 500 mM NaCl, 5% v/v glycerol, 20 mM imidazole) supplemented with EDTA-free protease inhibitor (Roche Diagnostics Ltd.) and 0.2 mg/ml lysozyme.

Cells were lysed using sonication for 3 min on ice using 2 s pulses. All purification steps were performed at 4°C on an ÄKTAxpress purifier (GE Healthcare). His-tagged enzyme was purified using a HisTrap FF 5 ml column (GE Healthcare) equilibrated with Buffer A. The sample was loaded at a flow rate of 4 ml/min and step-eluted with Buffer B (50 mM Tris-HCl pH 8, 50 mM glycine, 500 mM NaCl, 5% glycerol, 500 mM imidazole). Eluted protein was subjected to further purification on a Superdex Hiload 26/60 S75 gel filtration column (GE Healthcare) at a flow rate of 3.2 ml/min using Buffer C (20 mM Hepes pH 7.5, 150 mM NaCl) and collected into 8 ml fractions. After analysis by SDS-PAGE, those fractions containing no traces of other contaminating proteins were pooled and dialyzed in Buffer C (50 mM phosphate pH 7.6, 100 mM NaCl) and concentrated. Protein concentration was measured at 280 nm with Nanodrop according to the manufacturer's software. Purified proteins were divided in 20 μl aliquots, fast-frozen in liquid nitrogen and stored at −20 °C.

CYP71D1V1 enzyme was expressed in the *Saccharomyces cerevisiae* WAT11 cells[32] grown in synthetic complete without histidine medium (SC–His) plus 2% Glucose. A starter culture was grown for 48 h at 30 °C in 100 mL of SC–His media supplemented with 2% Glucose. A 1:100 dilution in fresh SC-His (4 × 1 l) media supplemented with 2% Glucose was prepared and allowed to grow at 30°C 24 h. Yeast cells were collected by centrifugation and resuspended in SC-His (4 × 1 l) media and protein production was induced by addition of 2% galactose and the cultures were shaken at 30°C overnight. The following day (20 h later), cells were collected by centrifugation, lysed by French Disruption Cell Press (one shot at 25 kPSI) in Buffer D (50 mM Tris, pH 7.4, 1 mM EDTA, 0.6 M sorbitol). After centrifugation at 35,000×g for 10 min and precipitation of cell debris, the supernatant was centrifuged at 100,000×g for 1 h. The pellet was resuspended in Buffer E (50 mM Tris, pH 7.4, 1 mM EDTA, 20 % glycerol). In total, from 4 l of yeast culture, a yield of 28 ml of microsomal protein resulted. Purified microsomal protein was divided in 500 μl aliquots, fast-frozen in liquid nitrogen and stored at −80 °C.

**Enzyme assays**. Purified GS1 or GS2 and purified SGD[17] were used in all assays. CYP71D1V1 was used as a microsome preparation as described above. Strictosidine was purified by preparative reverse phase HPLC and quantified using [1]H NMR as previously described[17]. The strictosidine aglycone substrate was generated by deglycosylating strictosidine (100 or 300 μM) by the addition of purified SGD in the presence of 50 mM phosphate buffer (pH 7.0) at 37 °C for 15 min. The reactions were started by the addition of GS1 or GS2 enzymes (5 μM), NADPH (5 mM) and microsomal protein containing GO (CYP71D1V1) (25 μl) at 37 °C in a total volume of 100 μl with light shaking. Caffeine (50 μM) was used as internal standard. All reactions were performed in quadruplicate. Aliquots of the reaction mixtures (10 μl) were sampled 2, 5, 15, 30, 45 and 60 min after addition of the GS and GO (CYP71D1V1) enzyme. The reactions were stopped by the addition of 10 μl of 100% MeOH. Samples were diluted further 1:5 in mobile phase ($H_2O$ + 0.1%

formic acid) and centrifuged for 10 min at 4000×g before UPLC-MS injection (1 µl). The activity of enzymes was measured by UPLC-MS.

UPLC-MS chromatography was performed on a BEH Shield RP18 (50 × 2.1 mm; 1.7 µm) column (Waters). The solvents used were $H_2O$ + 0.1% formic acid as Solvent A1 and 100% acetonitrile as Solvent B1, with a flow rate of 0.6 ml/min. Injection volume was 1 µl. The gradient profile was 0 min, 5% B1; from 0 to 3.5 min, linear gradient at 35% B1; from 3.5 min to 3.75 min, linear gradient to 100% B1; wash at 100% B1 for 1 min; from 4.75 min to 6 min, back to 5% B1 for 1 min to re-equilibrate the column.

Mass spectrometry detection was performed on a Waters Xevo TQ-S mass spectrometer (Milford, MA, USA) equipped with an electrospray (ESI) source. Capillary voltage was 2.5 kV in positive mode; the source was kept at 150 °C; desolvation temperature was 500 °C; cone gas flow, 50 l/h; and desolvation gas flow, 900 l/h. Unit resolution was applied to each quadrupole.

Targeted methods for each compound were developed using either commercial standards (caffeine, ajmalicine, serpentine, catharanthine, vindoline, vindorosine, loganic acid, tetrahydroalstonine, and yohimbine were purchased from Sigma-Aldrich and tabersonine from Avachem Scientific) or enzymatically produced compounds isolated as described here or as previously reported (strictosidine[17] and isositsirikine, this work). Akuammicine was synthesized and provided as a gift[26]. Flow injections of each individual compound were used to optimize the MRM conditions. This was done automatically using the Waters Intellistart software. A minimum dwell time of 25 ms was applied to each MRM transition. Four transitions were used to monitor the elution of tetrahydroalstonine and ajmalicine: $m/z$ 353.2 > 117.0 (Cone 50, Collision 46), $m/z$ 353.2 > 144.0 (Cone 50, Collision 26), $m/z$ 353.2 > 210.1 (Cone 50, Collision 20) and $m/z$ 353.2 > 222.0 (Cone 50, Collision 20). Transition $m/z$ 353.2 > 144.0 was used for quantification of these two compounds. Transitions $m/z$ 195.2 > 110.1 (cone 36, Collision 22) and $m/z$ 195.2 > 138.2 (cone 36, Collision 18) were used for caffeine; transitions $m/z$ 351.3 > 144.1 (cone 28, Collision 24) and $m/z$ 351.3 > 170.2 (cone 28, Collision 22) were used for deglycosylated strictosidine; transitions $m/z$ 531.3 > 144.1 (cone 32, Collision 36) and $m/z$ 351.3 > 352.2 (cone 32, Collision 24) were used for detection of strictosidine. Transitions $m/z$ 355.3 > 117.1 (Cone 48, Collision 42), $m/z$ 355.3 > 144.1 (Cone 48, Collision 26), $m/z$ 355.3 > 212.2 (Cone 48, Collision 18) and $m/z$ 355.3 > 224.2 (Cone 48, Collision 18) were used for detection of isositsirikine, yohimbine. Transitions $m/z$ 323.2 > 182.0 (Cone 50, Collision 36), $m/z$ 323.2 > 234.1 (Cone 50, Collision 34), $m/z$ 323.2 > 291.3 (Cone 50, Collision 22) were used for detection of akuammicine. Peak areas were calculated using the Waters MassLynx software and normalized.

The apparent rate under saturating substrate concentrations (observed $k_{cat}$) for GS1 was measured using NADPH absorbance using strictosidine (100 µM), NADPH (100 µM) and GS1 (0.83 µM) in an assay previously reported[17]. The apparent $k_{cat}$ was 0.04 ± 0.007 s$^{-1}$. Initial rates for GO can be calculated from the data presented in Fig. 3a: 9.73 nmol l$^{-1}$ s$^{-1}$ (100 µM concentration of strictosidine, 5 µM GS1, 5 mM NADPH) and 7.99 nmol l$^{-1}$ s$^{-1}$ (300 µM strictosidine, 5 µM GS1, 5 mM NADPH). The specific activity for GO was not calculated since it was used as a crude microsomal preparation (as is standard for microsomal cytochrome P450s).

**Chiral column liquid chromatography.** Synthetic akuammicine and enzymatically produced akuammicine were subjected to chiral resolution analysis by LCMS. Chromatography was performed on a Lux i-Cellulose 5 (150 × 3.0 mm, 3 µm) column (Phenomenex) under isocratic conditions using 60% 10 mM ammonium acetate and 40% acetonitrile for 10 min with a flow rate of 0.6 ml/min. Synthetic akuammicine (1 µl of 125 ng/ml solution) and 1 µl of enzyme assay were injected. Mass spectrometry detection was performed on a Waters Xevo TQ-S mass spectrometer using the transitions $m/z$ 323.2 > 182.0, $m/z$ 323.2 > 234.1, $m/z$ 323.2 > 291.3 for detection of akuammicine.

**Formaldehyde release detection assay.** Release of formaldehyde as result of rearrangement catalyzed by GO (CYP71D1V1) was monitored using a fluorescence-based Nash assay as previously reported[33]. Nash reagent was prepared by adding 0.3 ml of acetic acid and 0.2 ml of acetylacetone to 100 ml of 2 M ammonium acetate. Enzyme assays were performed in a total volume of 200 µl, using 300 µM strictosidine aglycone substrate, 5 µM GS1, 5 mM NADPH and various quantities of GO (CYP71D1V1) microsomes ranging from 5 to 100 µl in 50 mM phosphate buffer (pH 7.0). After 20 min, enzyme assays were centrifuged for 5 min at 3130×g for 5 min, 100 µl of enzyme assay were quenched with two volumes of Nash reagent and incubated at 60 °C for 10 min to convert formaldehyde to diacetyldihydrolutidine, which was detected by fluorescence using a BMG Labtech CLARIOstar microplate reader at $\lambda_{ex}$ = 412 nm and $\lambda_{em}$ = 505 nm. A calibration curve using formaldehyde (Sigma-Aldrich) from 0–50 µM was made for quantification. Peak areas were calculated using the BMG Labtech CLARIOstar software and normalized. Aliquots of the reaction mixtures (10 µl) after centrifugation were quenched by the addition of 10 µl of 100% MeOH. Samples were diluted further 1:5 in mobile phase ($H_2O$ + 0.1% formic acid) and centrifuged for 10 min at 4000×g before UPLC-MS injection (1 µl) and the activity of enzymes was measured by UPLC-MS as described above. Peak areas were calculated using the Waters MassLynx software and normalized.

**Large-scale enzyme assay.** For full characterization of GS1 and GO (CYP71D1V1) products, two large-scale enzyme assays were performed. For the characterization of the compounds (16R)-E-isositsirikine (1) and (16R)-Z-isositsirikine (2) produced by GS1, 9.2 mg of strictosidine prepared as described previously[17] were dissolved in 50 mL of 50 mM phosphate buffer (pH = 7.0). To generate strictosidine aglycone as substrate, 5 nM of SGD enzyme was added and incubated at 37 °C for 20 min. The consumption of strictosidine was monitored by flow injection MS performed on an Advion express-ion Compact Mass Spectrometer. NADPH (25 mg) and aliquots of GS1 to final concentration 5 µM were added to the reaction mixture. A NADPH-generation/regeneration system consisting of glucose, glucose dehydrogenase (Sigma-Aldrich) was used additionally. The reaction was monitored by flow injection MS at $m/z$ 355.3 and after 3 h was stopped by addition of 10 mL MeOH. The reaction mixture was centrifuged for 10 min at 3,130 g. The supernatant was loaded onto an SPE cartridge (RP18; 10 g; 60 ml), the SPE cartridge washed with 2 volumes of water and the enzyme reaction product was eluted with 1 volume of MeOH. The methanol SPE fraction was concentrated to 4 ml, filtered and products of GS1 were purified by preparative HPLC (details of the method are below).

For the characterization of compound 3 (akuammicine) produced by GS1 and GO (CYP71D1V1), 8.4 mg of strictosidine were dissolved in 50 ml of 50 mM phosphate buffer (pH = 7.0). To generate strictosidine aglycone as substrate, 5 nM of SGD enzyme were added and incubated at 37 °C for 20 min. The consumption of strictosidine was monitored by flow injection MS performed on an Advion express-ion Compact Mass Spectrometer. NADPH (25 mg), aliquots of GS1 to final concentration 5 µM and 12.5 ml of microsomal protein containing GO (CYP71D1V1) were added to reaction mixture. A NADPH-generation/regeneration system consisting of glucose, glucose dehydrogenase (Sigma-Aldrich) was also used. The reaction was monitored by flow injection MS at $m/z$ 323.2 and after 3 h was stopped by addition of 10 ml MeOH. The reaction mixture was centrifuged for 10 min at 3130×g. The supernatant was loaded onto an SPE cartridge (RP18; 10×g; 60 ml), the SPE cartridge was washed with 2 volumes of water and enzyme reaction product was eluted with 1 volume of MeOH. The methanol SPE fraction was concentrated to 4 ml, filtered and products of enzymatic reaction were purified by preparative HPLC.

Preparative HPLC was performed on a Thermo Dionex Ultimate 3000 chromatography apparatus (Ultimate 3000 Pump, Ultimate 3000 Various Wavelength Detector) using a Waters XBridge BEH C18 5 µ 10 × 250 OBD.). The solvents used ammonium hydroxide ($NH_4OH$) as Solvent A3 and 100% acetonitrile as Solvent B3, with a flow rate of 5.0 ml/min. Injection volume was 500 µl. For the isolation the gradient profile was 0 min, 30% B3; from 0 to 15 min, linear gradient to 95% B3; wash at 95% B3 for 2.5 min; from 17.5 min to 18 min, back to 30% B3 for 6 min to re-equilibrate the column. Elution of (16 R)-E-isositsirikine (1) and (16 R)-Z-isositsirikine (2) was monitored at 225 nm, while elution of akuammicine (3) was monitored at 328 nm. (16 R)-E-isositsirikine (1) was collected at 7.3 min, (16 R)-Z-isositsirikine (2) at 8.6 min and akuammicine (3) at 9.7 min. The collected fractions were analyzed on a Shimadzu LCMS-IT-TOF Mass Spectrometer. The chromatographic separation was carried out on a Phenomenex Kinetex column 2.6 u XB-C18 100 Å (100 × 2.10 mm, 2.6 µm), and the binary solvent system consisted of Solvent A4, $H_2O$ + 0.1% formic acid, and Solvent B4, acetonitrile, at a constant flow rate 600 µl/min. The LC gradient began with 10% Solvent B4 and linearly increased to 30% Solvent B4 in 5 min, then increased to 90% B3 in 1 min, held for 1.5 min and brought back to 10% Solvent B4. The fractions containing the compound of interest were combined, dried and used further for compound characterization. In total, 2.3 mg of (16 R)-E-isositsirikine (1)[24], ≈ 400 µg of (16 R)-Z-isositsirikine (2)[24] and ≈ 600 µg of akuammicine (3)[26] were isolated.

**Compound characterization.** High-resolution electrospray ionization mass spectrometry spectra and UV-Vis spectra were measured with a Shimadzu IT-TOF mass spectrometer as described above. NMR spectra (¹H NMR, ¹³C NMR) were acquired using a Bruker Avance III 400 NMR spectrometer equipped with a BBFO plus 5 mm probe, operating at 400 MHz for ¹H and 100 MHz for ¹³C. The residual ¹H- and ¹³C NMR signals of $CD_3Cl$ (δ 7.26 for ¹H and δ 77.36 for ¹³C) were used as internal chemical shift references. The number of scans depended on the concentration of the sample. The ¹H NMR spectra were compared with those of standards and literature data[24, 26]. For NMR, mass spectrometry and spectroscopic data, see Supplementary Figs 5–23.

**(16R)-E-isositsirikine (1).** ¹H-NMR (400 MHz, CDCl₃): δ 8.66 p.p.m. (s, br, 1 H),), 7.48 (d, br, $J$ = 7.6 Hz, 1 H), 7.38 (d, br, $J$ = 7.9 Hz, 1 H), 7.16 (ddd, $J$ = 7.9 Hz, $J$ = 7.2 Hz, $J$ = 1.1 Hz, 1 H), 7.10 (ddd, $J$ = 7.6 Hz, $J$ = 7.2 Hz, $J$ = 1.1 Hz, 1 H), 5.63 (q, $J$ = 6.8 Hz, 1 H), 4.32 (m, br, 1 H), 3.82 (s, 3 H), 3.57 (dd $J$ = 11.3 Hz, $J$ = 5.8 Hz, 1 H), 3.54 (d $J$ = 12.3 Hz, 1 H), 3.51 (dd $J$ = 11.3 Hz, $J$ = 4.7 Hz, 1 H), 3.28 (ddd, $J$ = 13.0 Hz, $J$ = 5.7 Hz, $J$ = 0.9 Hz, 1 H), 3.19–3.09 (m, 2 H), 3.05–2.96 (m, 1 H), 2.94 (d, $J$ = 12.2 Hz, 1 H), 2.65 (ddd, $J$ = 14.5 Hz, $J$ = 4.5 Hz, $J$ = 1.0 Hz, 1 H), 2.51 (ddd, $J$ = 11.4 Hz, $J$ = 7.8 Hz $J$ = 5.0 Hz, 1 H), 2.29–2.15 (m, 2 H), 1.68 (dd, $J$ = 6.9 Hz, $J$ = 1.7 Hz 3 H); ¹³C-NMR (125 MHz, CDCl₃): δ 175.6, 136.3, 134.1, 133.7, 127.8, 123.6, 121.7, 119.7, 118.1, 111.4, 108.0, 62.3, 52.9, 52.5, 52.4, 51.6, 49.7, 32.7, 30.5, 17.8, 13.5; UV/vis: $\lambda_{max}$ 225, 280 nm; HRMS ($m/z$): [M + H]⁺ calcd for $C_{21}H_{27}N_2O_3$, 355.2016; found, 355.2020; analysis.

**(16R)-Z-isositsirikine (2)**. ¹H-NMR (400 MHz, CDCl₃): δ 7.88 p.p.m. (s, br, 1 H), 7.46 (d, br, J = 7.6 Hz, 1 H), 7.32 (d, br, J = 7.9 Hz, 1 H), 7.14 (ddd, J = 7.9 Hz, J = 7.2 Hz, J = 1.1 Hz, 1 H), 7.08 (ddd, J = 7.6 Hz, J = 7.2 Hz, J = 1.1 Hz, 1 H), 5.46 (q, J = 6.8 Hz, 1 H), 3.91 (dd J = 11.3 Hz, J = 7.8 Hz, 1 H), 3.81 (dd J = 11.3 Hz, J = 4.5 Hz, 1 H), 3.78 (d J = 12.7 Hz, 1 H), 3.74 (s, 3 H), 3.63 (d, br, J = 10.5 Hz, 1 H), 3.17 (ddd, J = 10.8 Hz, J = 5.5 Hz, J = 1.0 Hz, 1 H), 3.05–2.93 (m, 2 H), 2.90 (d, J = 12.7 Hz, 1 H), 2.78–2.67 (m, 2 H), 2.67–2.58 (m, 1 H), 2.15 (ddd, J = 12.5 Hz, J = 4.5 Hz, J = 4.0 Hz, 1 H), 1.72 (d, J = 6.8 Hz, 3 H), 1.70–1.65 (m, 1 H); ¹³C-NMR (125 MHz, CDCl₃) (extracted from ¹H – ¹³C HSQC): δ 121.7, 119.7, 119.1, 118.3, 111.2, 62.7, 58.7, 54.7, 52.6, 52.0, 49.2, 41.1, 34.5, 21.2, 13.5 UV/vis: $\lambda_{max}$ 225, 279 nm; HRMS (m/z): [M + H]⁺ calcd for $C_{21}H_{27}N_2O_3$, 355.2016; found, 355.2014; analysis.

**Akuammicine (3)**. ¹H-NMR (600 MHz, CDCl₃): δ 9.00 p.p.m. (s, br, 1 H), 7.24 (d, br, J = 7.6 Hz, 1 H), 7.15 (ddd, J = 7.6 Hz, J = 7.6 Hz, J = 1.0 Hz, 1 H), 6.89 (ddd, J = 7.6 Hz, J = 7.6 Hz, J = 1.0 Hz, 1 H), 6.82 (d, br, J = 7.6 Hz, 1 H), 5.35 (q, br, J = 6.2 Hz, 1 H), 4.03 (m, br, 1 H), 3.94 (m, br, 1 H), 3.89 (d, br, J = 14.9 Hz, 1 H), 3.28 (m, 1 H), 3.02 (dd, J = 12.4 Hz, J = 6.7 Hz, 1 H), 2.95 (d, J = 15.0 Hz, 1 H), 2.51 (ddd, J = 12.6 Hz, J = 12.6 Hz, J = 6.8 Hz, 1 H), 2.42 (ddd, J = 13.8 Hz, J = 4.0 Hz, J = 2.3 Hz, 1 H), 1.82 (dd, J = 12.3 Hz, J = 5.5 Hz, 1 H), 1.60 (d, J = 6.9 Hz, 3 H), 1.30 (ddd, J = 13.8 Hz, J = 2.8 Hz, J = 2.8 Hz, 1 H); ¹³C-NMR (150 MHz, CDCl₃) (extracted from ¹H – ¹³C HSQC & ¹H – ¹³C HMBC): δ 167.8, 143.1, 139.3, 136.2, 127.8, 121.1, 120.9, 120.3, 109.5, 61.7, 57.4, 56.9, 56.2, 51.0, 46.1, 30.8, 29.7, 12.9. UV/vis: $\lambda_{max}$ 204, 225, 293, 328 nm; HRMS (m/z): [M + H]⁺ calcd for $C_{20}H_{23}N_2O_2$, 323.1754; found, 323.1760; analysis.

**Virus–induced gene silencing**. The GS1 silencing fragment was amplified with primers 5′-GGCGCGAUGTGTTTGCAATTTCGATATGG-3′ and 5′-GGTTGCGAUAUAGGATCGTTCCCCTTG-3′, to give a gene fragment of 266 bp. However, the very high nucleotide sequence identity (95%) of GS2 to GS1 means that co-silencing of these two genes is unavoidable. The resulting fragment, when subjected to a tnBLAST search against the C. roseus transcriptome at http://medicinalplantgenomics.msu.edu, did not show consecutive 22 bp duplication (minimal bp sequence needed for successful silencing) to any other gene, suggesting that cross-silencing with any other gene is highly unlikely. This fragment was cloned into the pTRV2u vector as described[17] and was used to silence the GS in C. roseus seedlings. Leaves from the first two pairs to emerge following inoculation were harvested from eight plants transformed with the empty pTRV2u and pTRV2u-GS1. Similarly, the GO (CYP71D1V1) silencing fragment was amplified using primers 5′-CTGAGAGGATCCTACAGTATGGCCCGA-3′ and 3′-CTGA-GAGGATCCATCGTTAACAAGATGAGGAACCAAT-5′, to generate a 298 bp cDNA displaying low identity with other C. roseus transcripts to avoid potential gene cross-silencing. After cloning this gene fragment into pTRV2u to generate pTRV2u-71D1V1, gene silencing was achieved on C. roseus plantlets through the biolistic-mediated inoculation of viral vectors[34, 35]. Leaves from the first pair to develop post-transformation were harvested from eight plants transformed with the empty pTRV2u or pTRV2u-71D1V1. For both types of transformation, the collected leaves were frozen in liquid nitrogen, powdered using a pre-chilled mortar and pestle (leaves from GS1- silenced plants) or using a mixer mill (Retsch MM400 for leaves from GO (CYP71D1V1) silenced plants), and subjected to LCMS and qRT-PCR analysis. Each pair of plant leaves was analyzed separately, for a total of 8 biological replicates.

To assess the results of VIGS, the alkaloid content of silenced leaves was determined by LCMS. To comprehensively assess the global effect of silencing on C. roseus metabolism using the GS1 fragment, an untargeted metabolomics analysis by LCMS was performed. Leaves were weighed (8–30 mg) and collected into a fixed volume of methanol (200 μl) and incubated at 56 °C for 60 min. After a 30 min centrifugation step at 5,000 g, an aliquot of the supernatant (20 μl) was diluted to 400 μl with water and analyzed on a Shimadzu LCMS-IT-TOF Mass Spectrometer. The chromatographic separation was carried out on a Phenomenex Kinetex column 2.6 μ XB-C18 100 Å (100 × 2.10 mm, 2.6 μm), and the binary solvent system consisted of Solvent A5, H₂O + 0.1% formic acid, and Solvent B5, acetonitrile, at a constant flow rate 600 μl/min. The LC gradient began with 10% Solvent B5 and linearly increased to 30% Solvent B5 in 5 min, then increased to 90% B5 in 1 min, held for 1.5 min and brought back to 10% Solvent B5. Peak areas were calculated using the Shimadzu Profiling Solution software and normalized by leaf mass (fresh weight). Analysis revealed that the only significant metabolic change with the GS1 silencing fragment was the accumulation of heteroyohimbines. The diastereomers tetrahydroalstonine and ajmalicine, which are both naturally present in C. roseus seedlings, do not separate under these chromatographic conditions.

For targeted metabolomic analysis, the secondary metabolite content of GS silenced leaves was determined by a different chromatographic method. Leaf powder was weighed (1–6 mg), extracted with methanol (2 ml) and vortexed for 1 min. After a 10-min centrifugation step at 17,000 g, an aliquot of the supernatant (200 μl) was filtered through 0.2 μm PTFE filters and analyzed on a Waters Xevo TQ-MS. The chromatographic separation was carried out with an Acquity BEH C18 1.7 μm 2.1 × 50 mm column and the binary solvent system consisted of solvent A6, which is 0.1% NH₄OH and solvent B6, which is 0.1% NH₄OH. A linear gradient from 0% to 65% B6 in 17.5 min was applied for separation of the

compounds followed by an increase to 100% B6 at 18 min, a 2 min wash step and a re-equilibration at 0% B6 for 3 min before the next injection. The column was kept at 60 °C throughout the analysis and the flow rate was 0.6 ml/min. MS detection was performed in positive mode ESI. Capillary voltage was 3.0 kV; the source was kept at 150 °C; desolvation temperature was 500 °C; cone gas flow, 50 l/h and desolvation gas flow, 800 l/h. Unit resolution was applied to each quadrupole. Multiple Reactions Monitoring (MRM) signals were used for detection and quantification of caffeine, heteroyohimbine alkaloids, strictosidine, akuammicine as reported above.

For targeted metabolic analyses of the alkaloid content of the GO (CYP71D1V1)-silenced plants, alkaloids were extracted from lyophilized leaves by grinding tissues with a mixer mill (Restch, MM 400) during 3 min at the maximal frequency. The resulting powders were incubated in 1 ml methanol (containing 0.1% formic acid) under vigorous shaking during 1 h at 24 °C. After centrifugation (15,000×g, 15 min), supernatants were collected and used for quantification. Alkaloid quantifications were performed using an UPLC-MS chromatography system coupled to a SQD mass spectrometer equipped with an electrospray ionization (ESI) source controlled by Masslynx 4.1 software (Waters, Milford, MA). Analyte separation was performed on a Waters Acquity HSS T3 C18 column (150 × 2.1 mm, i.d. 1.8 μm) with a flow rate of 0.4 ml/min at 55 °C and the volume of injection was 5 μL. The following linear elution gradient was used: acetonitrile-water-formic acid from 10:90:0.1 to 60:40:0.1 over 18 min. The capillary and sample cone voltages were 3,000 V and 30 V, respectively. The cone and desolvation gas flow rates were 60 and 800 l/h. MS experiments were carried out in positive mode in the selected ion-monitoring mode using m/z 337 for catharanthine ([M + H]⁺, RT = 12.33 min), m/z 457 for vindoline ([M + H]⁺, RT = 14.69 min), m/z 427 for vindorosine ([M + H]⁺, RT = 15.03 min), m/z 353 for ajmalicine ([M + H]⁺, RT = 11.7 min), m/z 349 for serpentine ([M + H]⁺, RT = 13.01 min) and m/z 375 for loganic acid ([M-H]⁻, RT = 4.94 min). The acquired data was processed by the QuanLynx™ software (Waters, UK). Relative quantification was performed by correcting peak areas by sample masses.

Gene silencing was confirmed by qRT-PCR. For GS1-silenced plants, RNA extraction was performed using the RNeasy Plant Mini Kit (Qiagen). RNA (1 μg) was used to synthesize cDNA in 20 μl reactions using the iScript cDNA Synthesis Kit (Bio-Rad). The cDNA served as template for quantitative PCR performed using the CFX96 Real-Time PCR Detection System (Bio-Rad) using the SSO Advanced SYBR Green Supermix (Bio-Rad). Each reaction was performed in a total reaction volume of 20 μl containing an equal amount of cDNA, 0.25 mM forward and reverse primers, and 1x SsoAdvanced SYBRGreen Supermix (Bio-Rad). The reaction was initiated by a denaturation step at 95 °C for 10 min followed by 41 cycles at 95 °C for 15 s and 60 °C for 1 min. For GO (CYP71D1V1) silenced plants, RNA was extracted with the NucleoSpin RNA Plant kit (Macherey- Nagel) and 1 μg from each extraction was retro-transcribed using the RevertAid first strand cDNA synthesis kit (ThermoFischer Scientific) with random hexamers (5 μM) according to the manufacturer's instructions. Gene expression levels were monitored by quantitative PCR performed using the Dynamo ColorFlash probe qPCR kit (ThermoFischer Scientific) in a 15 μl final volume containing 6 μl diluted template cDNA and the forward and reverse primers (0.5 μM). Amplifications were performed on a CFX96 real-time SYBR system (Bio-Rad) using detection of SYBR green with the following conditions: 95 °C for 7 min, 40 cycles at 95 °C for 10 s and 60 °C for 40 s. Melting curves were used to determine the specificity of the amplifications. Relative quantification of gene expression was calculated according to the delta-delta cycle threshold method using the 40 S ribosomal protein S9 (RPS9). The primers 5′-TTGAGCCGTATCAGAAATGC-3′ and 5′-CCCTCATCA AGCAGACCATA-3′ were used for RPS9, and 5′-TACTGAAGTTATTGGGAT TGA-3′ and 5′-TTCAATGTATTTCCAATGTCA-3′ were used for GS1 and 5′-GCTGAGTTTATGTTGGCTGCTATGTT-3′ and 5′-ATAGTTGGCAAAGA CAGACTAATCGT-3′ for GO (CYP71D1V1). All primer pair efficiencies were between 98% and 108%, and the individual efficiency values were considered in the calculation of normalized relative expression, which was performed using the Gene Study feature of CFX Manager Software. All biological samples were measured in technical duplicates (GS1-silenced plants) or triplicates (GO (CYP71D1V1)-silenced plants).

**Transcript distribution analysis**. Epidermis-enriched fractions were generated according to a previously described procedure[36]. Briefly, a cotton swab was dipped into a carborundum powder (particle size < 300 grit, Fischer) and used to abrade both lower and upper epidermis layers of young C. roseus leaves. Abraded leaves were dipped in 4 ml of Trizol (Life Technologies) for 5 s in a 15 mL centrifuge tube. A total of 3 × 10 leaves were abraded and RNAs were extracted according to the manufacturer's protocol. The RNA pellet resulting from the isopropanol precipitation was washed with 70% ethanol and re-suspended in 100 μl of RNAse-free water. Excess sugars were removed by precipitation with 10% ethanol for 5 min on ice and centrifugation for 5 min at 15,000×g and 4 °C. The supernatant was recovered and precipitated with 0.5 volume of 3 M sodium acetate (pH 5.2) and 2.5 volumes of 100% ethanol. The pellet was washed with 70% ethanol and re-suspended with 20 μl of RNase-free water. Total RNAs from whole young leaves were also extracted with Trizol (Life Technologies) according to the manufacturer's protocol. RNA from both fractions was quantified using a NanoDrop® ND-1000 and 1 μg was retro-transcribed with the RevertAid First Strand cDNA Synthesis Kit according to provider's instructions (ThermosFisher Scientific). Gene expression

levels were measured by quantitative PCR run on a CFX96 Touch Real-Time PCR System (Bio-Rad). Each reaction was performed in a total reaction volume of 25 µl containing an equal amount of cDNAs (1/3 dilution), 1 × DyNAmo ColorFlash Probe qPCR Kit (Thermo Fisher Scientific), and 0.05 µM forward and reverse specific primers, qSTRfor 5′-CATAGCTCTGTGGGTATATTAGTGT-3′, qSTRrev 5′-CATAGCTCTGTGGGTATATTAGTGT-3′; qSGDfor 5′-CTTCGACAACTTC GAATGGAA-3′; qSGDrev 5′-CTTCTTGACTAACTCAACTAGT-3′; qHDSfor 5′- GTCCCTTACTGAACCTCCAGAG-3′; qHDSrev 5′-AATCACCTGTCCTGC GTTGG-3′; qRPS9for 5′- TTACAAGTCCCTTCGGTGGT-3′, qRPS9rev 5′-TGC TTATTCTTCATCCTCTTCATC-3′: qGS1for 5′-TTGACTATCTCAGCACTGCT ATGGA-3′, qGS1rev 5′-GAACCAAGATGAAGACGGCACCTAA-3′ and qCYP71D1V1for 5′-ATGGATTTCTATGGAAGTAACTTTGAA-3′ and qCYP71D1V1rev 5′-AGTCCTCAGTCAAATCAATTTCTTCT-3′. The amplification program was 95 °C for 7 min (polymerase heat activation), followed by 40 cycles containing 2 steps, 95 °C for 10 sec and 60 °C for 40 sec. Quantification of transcript copy number was performed with calibration curves and normalization with the RPS9 reference gene and expressed relative to the amount of transcript measured in the whole leaf fraction. All amplifications were performed in triplicate and repeated at least on two independent biological repeats. Expression of hydroxymethylbutenyl 4-diphosphate synthase (HDS) known to accumulate within internal phloem associated parenchyma was used as a control[37].

**Subcellular localization**. Determination of the subcellular localization of GS1 and GO (CYP71D1V1) was performed by subcloning the full length GS1 and CYP71D1V1 coding sequences into the pSCA-cassette vector as YFP fusions. To avoid masking of the potential N-terminal transmembrane helix, CYP71D1V1 was only expressed as N-terminal fusion with YFP (GO (CYP71D1V1)-YFP) while GS1 was expressed as both N-terminal or C-terminal fusions (GS1-YFP and YFP-GS1). Amplifications were achieved with primers 5′-CTGAGAACTAGTATGGCCG GAGAAACAACCAAAC-3′ and 5′-CTGAGAACTAGTTTCCTCAAATTTCAAT GTATTTCCAATG-3′ for GS1; and 5′-CTGAGAACTAGTATGGAGTTTTCTT TCTCCTCACCAG-3′ and 5′-CTGAGAACTAGTATCGTTAACAAGATGAG GAACCAATT-3′ for CYP71D1V1. The nuclear (nucleus-CFP), nucleocytosolic (CFP) and endoplasmic reticulum (ER-CFP) markers were described previously[17]. Transient transformation of C. roseus cells by particle bombardment and fluorescence imaging were performed following the procedures previously described[35, 38]. C. roseus cells were bombarded with DNA-coated gold particles (1 µm) and 1,100 psi rupture disc at a stopping-screen-to-target distance of 6 cm, using the Bio-Rad PDS1000/He system and 400 ng of each plasmid per transformation. Cells were cultivated for 16 h to 38 h before being harvested and observed. The subcellular localization was determined using an Olympus BX-51 epifluorescence microscope equipped with an Olympus DP-71 digital camera and a combination of YFP and CFP filters.

**Data availability**. Sequence data for GS1 (KF302079.1), GS2 (KF302078.1). and GO (JN613015) have been deposited in GenBank as described in the manuscript. Transcriptome data used in this study has already been published as referenced in the manuscript (see also, http://medicinalplantgenomics.msu.edu/). Data supporting the findings of this study are available within the article and its supplementary information files and from the corresponding author upon reasonable request.

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

## Acknowledgements

This work was supported by grants from the European Research Council (311363), BBSRC (BB/J004561/1) (S.E.O.) and from the Région Centre-Val de Loire, France (ABISAL grant and BioPROPHARM project –ARD2020) (V.C.). J.F. gratefully acknowledges DFG postdoctoral funding (FR 3720/1-1) and T.T.-T.D. an EMBO Long trem Fellowship (ALTF 239-2015). We gratefully acknowledge the generous gift of akuammicine from Chris Vanderwal (UC Irvine).

## Author contributions

E.C.T., I.C., S.E.O., and V.C. designed the experiments and wrote the manuscript. E.C.T. characterized GS/GO in vitro, performed GS silencing, and performed all product characterization. J.F. and T.-T.T.D. contributed to in vitro enzyme assays. I.C., T.D.d.B., and V.C. identified GO, performed GO silencing, and GO silencing results and contributed to GS/GO in vitro assays and performed localization. I.C., E.C.T., and V.C. identified the link between GS in vitro reactions. A.K.S. contributed to identification and characterization of GS1 and T.D.d.B., A.O., A.L., F.L., and M.C. contributed to identification and characterization of GO.

## Additional information

**Competing interests:** The authors declare no competing financial interests.

