## [Peer Review File · Nature Communications]

Reviewers' comments:

Reviewer #1 (Remarks to the Author):

The manuscript by Tatsis et al. purports to have discovered that a particular combination of enzymes is essential for the biosynthesis of a key intermediate in monoterpene alkaloid biosynthesis. More specifically, the authors have provided a couple of different experimental approaches demonstrating that the combined activity of strictosidine glucosidase, a dehydrogenase (also known as geissoschizine synthase), and a P450 providing for an oxidative rearrangement yield akuammicine, a precursor of several different families of the monoterpene alkaloids. The manuscript is focused and written well, and the work is well done, especially the chemical identifications. While I agree that the authors have uncovered a possible mechanism for the generation of akuammicine, I'm not sure I agree with the intimation that the 3 enzyme sequence provides for streamlined system the authors would like us to conjure up in our heads.

First, all the information seems to be qualitative. Knockdown of the GO mRNA in the VIGS treated *C. roseus* plants results in a decreased accumulation of several monoterpenes and the appearance of one of the isositsirikine isomers, the E form. Naively, I would anticipate a quantitative relationship. The decreased monoterpene content should be matched by the increase in the isositsirikine level. Actually, one might also wonder about the level of strictosidine and its deglycosylated form. Regardless, I'm left wondering if the decrease in monoterpene alkaloids is matched by an equal increase in the level of isositsirikine or just a smidgeon. The real kickers here are two-fold: 1. "we noted in some replicates of CYP71DV1 silencing experiments, a compound...isositsirikin accumulated" – does this mean the result is not consistently seen?; 2. Why isn't there a parallel experiment for suppressing the 2 GS genes?

A somewhat similar confusion surrounds the in vitro experiments as well. When strictosidine aglycone (is this 4,21-dehydrogeissoschizine? – I'm guessing not by the way the text is written, but then later on aglycone is equated with dehydrogeissoschizine) is incubated with GS1, there are significant peaks on the chromatogram, which decrease when GO is added. I'd like to know how much of the starting substrate (strictosidine aglycone) is consumed, how much isositsirikine is produced with GS alone, and how much akuammicine is generated when the incubations include GS and GO. Why do I even mention this concern? Well, there seems to be a quantitative difference for the GS2 reactions that doesn't appear to add up. There are possible explanations for all this, but these cannot really be provided with some additional information.

Do I believe the authors have mapped a segment of the monoterpene alkaloid biosynthetic pathway from strictosidine to akuammicine? Yes, but I'm also left wondering how significant, quantitatively, this pathway is and if there are other possible ways/genes that could be contributing to this? I suppose the authors deserve a chance to amend the current manuscript to address my concerns.

On a more trivial side of things, the authors may also wish to provide citations for sentences like that referencing the opioid receptor agonist. There are a few such sentences scattered throughout.

Reviewer #2 (Remarks to the Author):

This is a very well written paper and the experiments are carefully conducted, presented and interpreted. However, the finding that when the plant enzyme(s) are expressed alone produce different products in eg *E. coli* or even plant cell system, then when the pathway is complete or nearly complete is not significantly novel.

Reviewer #3 (Remarks to the Author):

The manuscript by Tatsis et al. reports on the identification and characterization of genes involved in the conversions from strictosidine to prekuammicine, which are precursors in the biosynthesis of monoterpene indole alkaloids (MIAs), a class of natural products that encompasses several essential pharmaceuticals. As the authors correctly state, the nature of these enzymatic had remained enigmatic, and the topic of this manuscript is therefore of high interest to those working on understanding chemical diversity in plants. The narrative is a pleasure to read. The experiments and documentation to support the mechanistic claims are of excellent quality. Below are a few suggestions for further improvements/clarifications:

- In the abstract it is claimed that prekuammicine is the biosynthetic precursor of hundreds of pharmaceutically useful compounds. While there is certainly potential for MIAs as pharmaceuticals, only a very small number has been taken through clinical trials and the statement therefore needs to be toned down.
- I would suggest to add vincristine as a pharmaceutically relevant end product of the MIA pathway both in the narrative and figures. Its structure differs only slightly from that of vinblastine (so the inclusion in figures is easily realized) but both are important commercial products.
- I would refrain from referring to Scheme 2 in the Introduction as it forestalls the major outcomes of the study. This scheme is already partially integrated into Figure 1 and could easily be fully integrated.
- It would be desirable to list the sources of authentic standards. It is not entirely clear based on the narrative if the authors have those in hand for all metabolites mentioned in the text and shown in figures (obviously with the exception of those for which a structural characterization is presented here).
- The authors should consider the inclusion of high resolution mass spectrometry data in the narrative of the Results section. Such data sets enable the calculation of an empirical formula, which is a valuable guide for structure elucidation. The Methods section provides accurate m/z data, so I am just asking for inclusion in the storyline.
- On page 5, please refer to an m/z value rather than "mass" as output of a mass spectrometric analysis.
- I think that the authors present solid evidence for a coupling of GS1/GS2 and CYP71D1V1 and a plausible hypothesis as to how the reactions could proceed to generate prekuammicine. The only part of the characterization that is missing is an investigation of the kinetic properties of the recombinant enzymes. I recognize the experimental challenges but maybe an attempt could be made.
- The conclusions regarding the localization of gene products, both intracellularly and at the tissue level, are also based on solid experimental evidence.

Reviewer #1:

...The manuscript is focused and written well, and the work is well done, especially the chemical identifications. While I agree that the authors have uncovered a possible mechanism for the generation of akuammicine, I'm not sure I agree with the intimation that the 3 enzyme sequence provides for streamlined system the authors would like us to conjure up in our heads.

We thank the reviewer for the thoughtful and positive remarks. We hope that the responses to the more specific points below will fully convince the reviewer that these three enzymes together produce akuammicine.

*First, all the information seems to be qualitative. Knockdown of the GO mRNA in the VIGS treated *C. roseus* plants results in a decreased accumulation of several monoterpenes and the appearance of one of the isositsirikine isomers, the E form. Naively, I would anticipate a quantitative relationship. The decreased monoterpene content should be matched by the increase in the isositsirikine level. Actually, one might also wonder about the level of strictosidine and its deglycosylated form. Regardless, I'm left wondering if the decrease in monoterpene alkaloids is matched by an equal increase in the level of isositsirikine or just a smidgeon.*

The reviewer is correct that our conclusions regarding the VIGS are qualitative. This is because the decrease in a metabolite is not quantitatively matched by the increase in its precursor in these “*in planta*” silencing experiments. There are many possible reasons why the decrease of product and increase of precursor may not add up: the accumulating compound could be degraded or modified by endogenous enzymes in the plant, or the compound could be transported to and from the silenced leaves to other parts of the plant. Strictosidine aglycone, for example, does not ever accumulate in the plant, presumably due to its reactivity. In VIGS, which is a transient silencing experiment, the level of silencing varies from replicate to replicate, and so any quantitative conclusions drawn from the levels of metabolite accumulation are even less valid.

However, we have now provided the normalized data for the levels of accumulation of isositsirikine in all silencing replicates in the new Supplementary Figure 1b so that the reader can see the data. Additionally, we have also provided a brief sentence on the limitations of VIGS and also highlight that the VIGS data, while statistically significant, should only be considered qualitatively. “Gene candidates identified were initially screened in *C. roseus* by virus induced gene silencing (VIGS), **a transient silencing system in which qualitative perturbations in the metabolic profile between silenced and control tissue provide clues to the metabolic function of the gene of interest.**”

The real kickers here are two-fold: 1. “we noted in some replicates of CYP71DV1 silencing experiments, a compound....isositsirikin accumulated” – does this mean the result is not consistently seen?

As mentioned in the response above, the reader can now see the levels of accumulation of *m/z* 355 in all silencing replicates in Supplementary Figure 1b. The compound is consistently

observed, though the levels do vary from replicate to replicate, which is expected for the VIGS technique.

2. Why isn't there a parallel experiment for suppressing the 2 GS genes?

GS was also subjected to VIGS, as discussed in the text and as shown in Supplementary Figure 2. These data are not as conclusive as those for GO: only strictosidine accumulation is changed (increased) to a statistically significant degree. All other metabolite changes measured with targeted and untargeted mass spectrometry were not statistically significant. The increased accumulation of strictosidine is strongly suggestive that GS is involved in the monoterpene indole alkaloid pathway in planta, but the lack of statistically significant decreases in downstream alkaloids could be indicative of redundancy: there could be another reductase that could be involved in this process. We have now explicitly stated this: "Since VIGS phenotypes can be attenuated by factors such as enzyme redundancy or molecule transport, we nevertheless assayed the protein *in vitro*."

A somewhat similar confusion surrounds the in vitro experiments as well. When strictosidine aglycone (is this 4,21-dehydrogeissoschizine? – I'm guessing not by the way the text is written, but then later on aglycone is equated with dehydrogeissoschizine) is incubated with GS1, there are significant peaks on the chromatogram, which decrease when GO is added. I'd like to know how much of the starting substrate (strictosidine aglycone) is consumed, how much isositsirikine is produced with GS alone, and how much akuammicine is generated when the incubations include GS and GO. Why do I even mention this concern? Well, there seems to be a quantitative difference for the GS2 reactions that doesn't appear to add up. There are possible explanations for all this, but these cannot really be provided with some additional information.

First, 4, 21-dehydrogeissoschizine is one of the isomers that strictosidine aglycone can form. We have tried to make this clearer by consistently specifying this in the text and figures.

Next, we think that the reviewer is asking if the yields of the enzymatic reactions are close to 100%: the answer is that they are not. In all reactions, we observe complete consumption of strictosidine with the glucosidase. We do not attempt to quantify the amounts of strictosidine aglycone, since this compound gives a broad peak or peaks (depending of LC conditions), and also crosslinks with any protein present in the sample (see Stavrinides, Chem Biol, 2015). We can estimate isolated yields of isositsirikine and akuammicine from the large-scale reactions, which are reported in the Methods, and which we now include in the main text. In the reaction containing strictosidine, SGD and GS1, yields of isositsirikine are 38% for the E isomer and 7% for the Z isomer. These yields are calculated from the starting concentration of strictosidine, and are measured by weighing the isolated compounds that are assumed to be 100% pure (so the reviewer needs to bear in mind there is certainly some error associated with these values, but they do provide a clear idea of the yields). In the reaction containing strictosidine, SGD, GS1 and GO, the isolated yield for akuammicine is 12.5%. We did not attempt to isolate the isositsirikine products in this reaction, but it is clear that less isositsirikine is being produced compared to reactions when GO is absent, and the total isolated product (akuammicine + isositsirikine) is far below

100%. The inefficiency of the reaction is in large part due to the fact that strictosidine aglycone is highly reactive, and a substantial amount is lost in the reaction due to crosslinking with the enzymes present (see Stavrinides, Chem Biol, 2015). Note that in a previous publication, where we reported other medium chain alcohol dehydrogenases that reduce strictosidine aglycone, the overall yields were also in the 30% range (Stavrinides, Tatsis et al., Nat Comms, 2016).

Non-isolated yields are shown for akuammicine in Figure 3 and Supplementary Figure 26. These data indicate that the total yield of akuammicine is between 15-20%, which is consistent with the isolated yield. These yields could be reliably quantified with the authentic standard of akuammicine provided by Vanderwal. Non-isolated yields were not calculated for isositsirikine since we had concerns about the accuracy of the concentration of the stock solution that was used for the standard curve, which was made from the small amounts of isolated isositsirikine.

The reviewer states “*there seems to be a quantitative difference for the GS2 reactions that doesn't appear to add up*”.

We think the reviewer is referring to the fact that the GS2 homolog is much less active. Both the amount of isositsirikine and akuammicine produced using this enzyme are much less compared to GS1. The reasons for this are not clear, though we think that the fact that GS2 expresses much less well in the heterologous E. coli expression system may contribute to this. Given the very low amounts of product formed, we did not attempt to quantitate these yields or rates.

Do I believe the authors have mapped a segment of the monoterpene alkaloid biosynthetic pathway from strictosidine to akuammicine? Yes, but I'm also left wondering how significant, quantitatively, this pathway is and if there are other possible ways/genes that could be contributing to this? I suppose the authors deserve a chance to amend the current manuscript to address my concerns.

It is always a challenge to definitively establish the physiological relevance of an enzyme, and the reviewer is correct that we cannot absolutely rule out other mechanisms of akuammicine formation in planta. Nevertheless, the VIGS data strongly suggest that GO is the major player involved in the biosynthesis of preakuammicine derived alkaloids. While, as described above, a strict quantitative relationship among the levels of gene silencing, product decrease and precursor increase cannot be established in an in vivo silencing system, the statistically significant decrease in the preakuammicine-derived alkaloids is a clear indication for the essential role of GO in this pathway. The silencing data for GS are less clear, a result that could be due to a number of factors, including the presence of reductases with redundant function. Therefore, we cannot rule out that there is another reductase that also carries out this catalytic function. We now say this explicitly in the manuscript. However, the relatively efficient in vitro production of the complex molecule akuammicine is unlikely to be the result of a non-specific or off-target enzyme reaction. (In this system, with the high reactivity/crosslinking of strictosidine aglycone, we consider 10-20% akuammicine yield to be relatively efficient.)

On a more trivial side of things, the authors may also wish to provide citations for sentences like that referencing the opioid receptor agonist. There are a few such sentences scattered throughout.

We have highlighted where we have added this and additional references in the discussion.

Reviewer #2:

This is a very well written paper and the experiments are carefully conducted, presented and interpreted. However, the finding that when the plant enzyme(s) are expressed alone produce different products in eg E. coli or even plant cell system, then when the pathway is complete or nearly complete is not significantly novel.

We agree that the statement that plant enzymes must work in concert to give the appropriate product has been observed before. However, it is an important lesson to keep in mind when elucidating complex plant pathways with unstable intermediates, which is why we emphasized the point. However, the main novelty of the manuscript is the unusual reaction and the elucidation of the biosynthesis of an important alkaloid intermediate.

To make sure that we don't over-emphasize the novelty of enzymes working in tandem, we changed "This discovery that these enzymes can function in combination..." to "This discovery of how these enzymes can function in combination..." in the abstract. In the discussion we changed "Importantly, this work has demonstrated that to unlock the chemical diversity of certain plant pathways..." to "Importantly, this work has highlighted that to unlock the chemical diversity of certain plant pathways..."

Reviewer #3:

In the abstract it is claimed that prekuammicine is the biosynthetic precursor of hundreds of pharmaceutically useful compounds. While there is certainly potential for MIAs as pharmaceuticals, only a very small number has been taken through clinical trials and the statement therefore needs to be toned down.

We agree with the reviewer and have replaced "pharmaceutically useful" with "biologically active" in the abstract and introduction.

I would suggest to add vincristine as a pharmaceutically relevant end product of the MIA pathway both in the narrative and figures. Its structure differs only slightly from that of vinblastine (so the inclusion in figures is easily realized) but both are important commercial products.

Vincristine has now been added to the text and Scheme 1.

I would refrain from referring to Scheme 2 in the Introduction as it forestalls the major outcomes of the study. This scheme is already partially integrated into Figure 1 and could easily be fully integrated.

We have combined Scheme 1 and Scheme 2 into Scheme 1a and Scheme 1b. Scheme 1a shows that preakuammicine is the precursor for many alkaloids. Scheme 1b shows the details of the proposed chemical transformation of strictosidine aglycone to preakuammicine. We have removed the enzyme names from Scheme 1b. We hope this is what the reviewer had in mind; we can change the scheme further if needed if we did not understand the request.

It would be desirable to list the sources of authentic standards. It is not entirely clear based on the narrative if the authors have those in hand for all metabolites mentioned in the text and shown in figures (obviously with the exception of those for which a structural characterization is presented here).

We have made sure that the source of all authentic standards is now included in the methods section. See the highlighted sections.

The authors should consider the inclusion of high resolution mass spectrometry data in the narrative of the Results section. Such data sets enable the calculation of an empirical formula, which is a valuable guide for structure elucidation. The Methods section provides accurate m/z data, so I am just asking for inclusion in the storyline.

We have added the HRMS data to the results: see highlighted text. We did not include the calculated values or molecular formula (these data are currently in the Methods), but we can also add these to the Results if it makes the text stronger.

On page 5, please refer to an m/z value rather than “mass” as output of a mass spectrometric analysis.

This is fixed.

I think that the authors present solid evidence for a coupling of GS1/GS2 and CYP71D1V1 and a plausible hypothesis as to how the reactions could proceed to generate preakuammicine. The only part of the characterization that is missing is an investigation of the kinetic properties of the recombinant enzymes. I recognize the experimental challenges but maybe an attempt could be made.

We agree with this request, but unfortunately, there are challenges associated with characterizing the steady state kinetic parameters of enzymes that act on strictosidine aglycone. The actual aglycone substrate is known to crosslink proteins and other biomolecules and it precipitates in solution during the course of the assay at higher concentrations. When we first reported the discovery of an alcohol dehydrogenase that acts on strictosidine aglycone to generate a heteroyohimbine alkaloid (Stavrinos et al. Chem. Biol. (2015) 22, 336–341), we attempted to obtain accurate steady state kinetic parameters from a very large number of kinetic experiments. The data consistently had large error bars at higher substrate concentrations, presumably because the substrate started to crosslink the enzyme. Visible precipitation is noted at substrate concentrations above 100 μ M. This

large error is why we ultimately decided not to measure the kinetic characteristics of the enzymes presented in this study.

As a way to provide more characterization while remaining realistic about the challenges of this system, we have measured the observed k_{cat} of GS1 (now reported in the methods). We measured this using an assay reported in (Stavrinides, Tatsis, Nat Comms, 2016), at a strictosidine concentration of 100 μM , which is well above saturating concentrations in ADH enzymes that use strictosidine aglycone as a substrate (Stavrinides, Tatsis, Nat Comms, 2016). The reader needs to bear in mind that this kinetic constant reflects the double reduction to form isositsirikine, and not the (presumably more relevant) single reduction to form geissoschizine. We established some initial rate data and concentration dependence for GO, as shown in Figure 3a and Supplementary Fig 26. The extracted rate values are now included in the Methods section of the manuscript. However, obtaining accurate kinetic values for GO is even more complicated since this enzyme assay uses GO as an unpurified microsomal preparation (standard for microsomal P450s) and this activity must be coupled to the single reduction activity of GS. Current efforts to produce geissoschizine (the GO substrate) by semi-synthesis are ongoing, and when we work out a way to isolate this compound in milligram quantities we hope to report more accurate V_{max} and K_{m} values for GO.

REVIEWERS' COMMENTS:

Reviewer #1 (Remarks to the Author):

The revised manuscript by Tatsis et al. satisfactorily addresses my previous questions and comments. The authors have definitely picked up on my concern that while they describe one means for the biosynthesis of preknammicine and akuammicine, the evidence that this is the only mechanism operating, especially in planta, is not iron clad. Assembling these enzymes in some sort of Synthetic Biology platform might yield the akuammicine product. But given the questions about qualitative versus quantitative evidence, I wanted them to make allowances for alternative pathways. They have done this. Making such allowances does not diminish the impact of the work, but could stimulate others to consider alternatives and thus contribute to an evolving story.

The addition of Supplementary Fig. 1B is informative and compelling. Even if the levels of isositsirikine are variable, this data nicely demonstrates the reproducibility of the experiment.

I also noted that the authors added some enzyme kinetic inferences to the methods section, but only for the GS reaction. They back off any such calculations for GO because of using "crude microsomes". It isn't possible to rectify this situation now, but much of P450 enzymology literature relies on this type of set-up and uses CO difference spectra to quantify the amount of properly folded P450. Thus, allowing for very quantitative enzymology. A few words and sentences in this section are corrupted and need to be repaired.

Shouldn't the Benayad reference in the legend of Fig. 3 have a citation number associated with it?

Reviewer #3 (Remarks to the Author):

The authors have addressed my concerns by updating the narrative and providing additional experimental data.

REVIEWERS' COMMENTS:

Reviewer #1 (Remarks to the Author):

The revised manuscript by Tatsis et al. satisfactorily addresses my previous questions and comments. The authors have definitely picked up on my concern that while they describe one means for the biosynthesis of preknammicine and akuammicine, the evidence that this is the only mechanism operating, especially in planta, is not iron clad. Assembling these enzymes in some sort of Synthetic Biology platform might yield the akuammicine product. But given the questions about qualitative versus quantitative evidence, I wanted them to make allowances for alternative pathways. They have done this. Making such allowances does not diminish the impact of the work, but could stimulate others to consider alternatives and thus contribute to an evolving story.

We appreciate the comments.

The addition of Supplementary Fig. 1B is informative and compelling. Even if the levels of isositsirikine are variable, this data nicely demonstrates the reproducibility of the experiment.

Again, this is appreciated.

I also noted that the authors added some enzyme kinetic inferences to the methods section, but only for the GS reaction. They back off any such calculations for GO because of using "crude microsomes". It isn't possible to rectify this situation now, but much of P450 enzymology literature relies on this type of set-up and uses CO difference spectra to quantify the amount of properly folded P450. Thus, allowing for very quantitative enzymology.

Agreed: but I think that this would not be worth doing until we obtain the geissoschizine substrate, since we are not sure if the reduction by GR is the rate determining step or not.

A few words and sentences in this section are corrupted and need to be repaired.

We have checked this.

Shouldn't the Benayad reference in the legend of Fig. 3 have a citation number associated with it?

We have added this.

Reviewer #3 (Remarks to the Author):

The authors have addressed my concerns by updating the narrative and providing additional experimental data.

We appreciate the comment.